# Mediterranean and MIND Dietary Patterns and Cognitive Performance in Multiple Sclerosis: A Cross-Sectional Analysis of the UK Multiple Sclerosis Register

**DOI:** 10.3390/nu17213326

**Published:** 2025-10-22

**Authors:** Maggie Yu, Steve Simpson-Yap, Annalaura Lerede, Richard Nicholas, Shelly Coe, Thanasis G. Tektonidis, Eduard Martinez Solsona, Rod Middleton, Yasmine Probst, Adam Hampshire, Elasma Milanzi, Guangqin Cui, Rebekah Allison Davenport, Sandra Neate, Mia Pisano, Harry Kirkland, Jeanette Reece

**Affiliations:** 1Neuroepidemiology Unit, Melbourne School of Population & Global Health, The University of Melbourne, Carlton, VIC 3010, Australia; 2MS Research Flagship, Menzies Institute for Medical Research, University of Tasmania, Hobart, TAS 7000, Australia; 3Florey Institute of Neuroscience and Mental Health, The University of Melbourne, Parkville, VIC 3052, Australia; 4Department of Brain Sciences, Faculty of Medicine, Imperial College London, London W12 0NN, UK; annalaura.lerede18@imperial.ac.uk (A.L.);; 5Centre for Neuroimaging Sciences, Institute of Psychiatry, Psychology & Neuroscience, King’s College London, London SE5 8AF, UK; 6Population Data Science, Swansea University Medical School, Swansea SA2 8PP, UK; 7School of Sport, Nutrition and Allied Health Professions, Oxford Institute for Applied Health Research, Oxford Brookes University, Oxford OX3 0BP, UKs224215148@deakin.edu.au (E.M.S.); 8Co-Centre for Sustainable Food Systems and The Institute for Global Food Security, Queen’s University Belfast, Northern Ireland, Belfast BT9 5DL, UK; 9Institute for Physical Activity and Nutrition (IPAN), School of Exercise and Nutrition Sciences, Deakin University, Burwood, VIC 3125, Australia; 10School of Medical, Indigenous and Health Sciences, University of Wollongong, Wollongong, NSW 2522, Australia; yasmine@uow.edu.au; 11Centre for Epidemiology and Biostatistics, Melbourne School of Population & Global Health, The University of Melbourne, Carlton, VIC 3010, Australia; 12Melbourne School of Psychological Sciences, The University of Melbourne, Parkville, VIC 3010, Australia

**Keywords:** multiple sclerosis, cognitive performance, cross-sectional study, mediterranean diet, MIND diet

## Abstract

Background: Multiple sclerosis (MS) is a chronic auto-immune neuroinflammatory disorder presenting as a range of systemic and neurological symptoms, including cognitive impairment. Emerging evidence suggests that diets targeting brain health—such as the Mediterranean (MED) and Mediterranean-DASH Intervention for Neurodegenerative Delay (MIND) diets—may improve cognitive function; however, studies examining their role in people living with MS are limited. Methods: We examined cross-sectional associations between diet and cognition data from 967 participants in the United Kingdom Multiple Sclerosis Register (UKMSR). Dietary pattern scores (alternate Mediterranean; aMED, and MIND) were derived from the 130-item EPIC-Norfolk food frequency questionnaire. Cognition was assessed using the MS-specific Cognitron-MS (C-MS) battery (13 tasks) and summarised as overall cognition (global G factor) and four domains (object memory, problem solving, information processing speed [IPS], and words memory). Cognitive outcomes were expressed as Deviation-from-Expected (DfE) scores standardised to demographic and device characteristics using external regression-based norms. Linear models were adjusted for total energy intake, MS phenotype, disease duration since diagnosis, and current disease-modifying therapy (DMT) use. Interactions tested moderation by MS phenotype (relapsing vs. progressive MS) and current DMT use (yes vs. no). Sensitivity analyses included within-domain multiple-comparison control, rank-based inverse-normal transformation, and winsorisation. Results: Greater alignment with aMED and MIND dietary patterns were associated with higher scores in specific cognitive domains but not in overall cognition. Higher aMED scores were associated most consistently with better IPS, while higher MIND scores were additionally associated with better words memory. In categorical models, participants with the middle or highest tertiles of aMED or MIND scores performed up to ~0.4 SD better on tasks of Verbal Analogies, Word Definitions, Simple Reaction Time, Words Memory Immediate, or Words Memory Delays compared with those in the lowest tertile. These findings were robust across sensitivity analyses. Stratified analyses showed differential cognitive performance and diet-cognition associations by MS phenotype and DMT use. Conclusions: Mediterranean and MIND dietary patterns showed modest cross-sectional associations with specific cognition domains, with differential cognitive performance in different subgroups according to MS phenotype and DMT use. Although causal inference is not possible, our findings indicate future MS-related dietary studies (longitudinal and/or randomised controlled trials) examining cognitive function domains across different MS subgroups are warranted.

## 1. Introduction

Multiple sclerosis (MS) is a chronic, immune-mediated disorder of the central nervous system (CNS) that affects more than 2.8 million people worldwide and approximately 150,000 in the United Kingdom [1,2]. The disease arises from dysregulated immune activation that triggers inflammation within the CNS, leading to demyelination, axonal injury, and progressive neurodegeneration [3]. These inflammatory and neurodegenerative processes underlie the diverse clinical manifestations of MS and contribute to its considerable heterogeneity in disease course, symptom profile, and long-term outcomes. The condition most commonly presents in early to mid-adulthood, with a peak onset between 20 and 40 years of age, and affects women approximately two to three times more frequently than men [4].

Cognitive impairment, a term used to describe deficits in different domains such as information processing, memory, executive function, reasoning, judgement, comprehension, attention [5], is recognised as a core clinical feature of MS, affecting approximately 40–70% of the population [6]. Moreover, people with a more advanced or progressive form of MS (e.g., primary progressive MS; PPMS) experience cognitive impairment at a higher frequency and severity, and across a wider range of cognitive functions and domains, than people with relapsing-remitting MS (RRMS), the most common phenotype of MS [7]. Notably, health-related quality of life (QoL), employment prospects and social engagement are frequently compromised due to cognitive impairment in people living with MS [8,9].

Despite the high prevalence and impact of cognitive impairment in people living with MS, the management of cognitive dysfunction remains insufficient [10]. Although evidence suggests newer disease-modifying therapies (DMTs) for MS aimed to reduce disability and progression, may reduce the incidence of cognitive impairment, their overall impact on cognition remains limited [11,12]. Further, in people with progressive MS, the effects of DMTs on cognitive outcomes are uncertain and variable [11,13,14,15]. In parallel, non-pharmacological strategies such as lifestyle modification (e.g., diet, physical activity, and stress reduction) may provide complementary cognitive benefits adjunct to routine clinical care [16,17,18,19].

Amongst the lifestyle behaviours explored in MS, dietary factors are of particular interest due to their potential to influence inflammation, oxidative stress, and gut–brain immune signalling, all of which are implicated in MS pathogenesis. As per a recent review, dietary components and their metabolites may affect these pathways by modulating immune activity, reducing oxidative damage, and shaping gut microbial composition, providing a biologically plausible basis for their relevance to MS [20]. In particular, certain dietary patterns are emerging as promising non-pharmacological strategies to support brain health and cognitive function [21,22]. For example, greater adherence to anti-inflammatory diets have been associated with a lower risk of cognitive decline and neurodegenerative disease in the general population [23,24]. The Mediterranean diet, rich in vegetables, fruits, whole grains, legumes, nuts, fish, and olive oil, has also been associated with inflammatory and oxidative pathways, and its dietary scores have been linked to better cognitive outcomes in people living with MS [22,25]; however, findings across related studies are inconsistent [26,27].

More recently, the neuroprotective diet, Mediterranean–DASH Intervention for Neurodegenerative Delay (MIND) diet, has shown promise in reducing risks of cognitive decline or impairment in individuals with Alzheimer’s and in those who have experienced stroke [28,29]. This diet is a hybrid of the Mediterranean diet and DASH (Dietary Approaches to Stop Hypertension) diet, with modifications to include foods considered beneficial for brain health (e.g., green leafy vegetables and berries). The MIND diet is thought to relate to biological processes relevant to MS, including oxidative stress and chronic inflammation, through its emphasis on polyphenol- and antioxidant-rich foods [30]. However, to the authors’ knowledge, only one study has examined MIND diet adherence and neurodegeneration outcomes in people living with MS [31]. In this cross-sectional study of people diagnosed with MS within the last 5 years (*N* = 180), higher MIND scores were associated with greater thalamic volume, a structure critical for cognitive function; however, actual cognitive testing was not performed in this study. In studies of other neurodegenerative diseases, higher MIND diet adherence was associated with slower cognitive decline and reduced Alzheimer’s incidence in older adults [28], as well as slower cognitive decline after stroke [32], and in late onset Parkinson’s disease [33]. However, in one large randomised controlled trial (RCT) examining MIND diet adherence in older adults at risk of Alzheimer’s disease, greater MIND diet adherence did not result in any improvements in cognitive function compared to the control group [34].

In the present study, we aimed to examine the relationship between Mediterranean and MIND diet scores and cognitive function in people living with MS. In particular, we performed cross-sectional analyses examining Mediterranean and MIND diet scores and cognitive performance using Cognitron from data extracted from participants within the United Kingdom Multiple Sclerosis Register (UKMSR). Due to the higher frequency and severity of cognitive impairment in people with progressive MS, we examined associations by MS phenotype (relapsing vs. progressive MS). As DMTs can potentially modify cognitive function, we also examined associations in participants currently taking DMTs compared with those not taking DMTs. Study findings may provide insights into self-management dietary strategies for people living with MS, specific for disease course and treatment regimens, and inform future mechanistic or interventional studies exploring how dietary patterns may influence cognitive health in MS.

## 2. Materials and Methods

### 2.1. Study Design and Participants

The UKMSR is a nationwide registry supported by National Health Service (NHS) clinical centres (Research Ethics Committee: 21/SW/0185, approval date 5 August 2021). Eligibility criteria for enrolling in the UKMSR includes a clinically confirmed diagnosis of MS, age over 18 years, and living in the United Kingdom [35]. The UKMSR collects longitudinal data through self-reported questionnaires covering demographics, clinical history, symptoms, lifestyle, and patient-reported outcomes (3-monthly from 2011 to 2018 and 6-monthly from 2018 onwards).

A large-scale online cognitive assessment was developed and delivered via the Cognitron assessment platform [36]. This cognitive assessment was integrated into the UKMSR’s digital infrastructure enabling it to be deployed to valid UKMSR participants (*N* = 19,188 with year of birth, sex, MS phenotype at diagnosis and disease duration since diagnosis data). The Cognitron tool comprised a sociodemographic questionnaire (collecting information such as ethnicity, language, education, handedness, and device use) and a series of online tasks designed to comprehensively assess cognitive domains commonly affected in people with MS.

This study comprised a cross-sectional analysis of participants from the UKMSR who completed both the Cognitron assessment (between November 2022 and January 2023) and the EPIC-Norfolk food frequency questionnaire (FFQ) [37] within the same 3-month period in 2022 (*N* = 993). Data linkage was achieved using unique participant identifiers, allowing integration of cognitive, dietary, and clinical data.

### 2.2. Demographics and Clinical Outcomes

The cognitive assessment included a self-reported sociodemographic questionnaire, collecting information on age, sex, dominant hand, education, ethnicity, first language, country of residence, and occupation. Disease-specific data were also queried; MS phenotype at diagnosis (categorised as “relapsing” MS—people living with benign MS, RRMS or people transitioning from RRMS to secondary progressive MS [SPMS] and “progressive” MS (people with SPMS or PPMS); disease duration (number of years since initial symptoms; the date of the cognitive assessment [36] and whether the participant was currently taking DMTs to manage their MS (yes/no).

### 2.3. Diet

Participants completed the 130-item EPIC-Norfolk FFQ [37] hosted by the UKMSR in 2022. This questionnaire collects self-reported information on the frequency and quantity of foods and drinks consumed over the previous 12 months, which is used to estimate habitual diet composition and adherence to dietary patterns. The FFQ enabled Mediterranean and MIND dietary pattern scores to be calculated using established criteria. Two dietary indices were applied: the alternative Mediterranean diet score (aMED) [38] and the MIND diet score [28]. 

The aMED score, adapted from the traditional Mediterranean diet scale [39], and developed to assess diet–disease associations in non-Mediterranean populations, was used to evaluate alignment with a Mediterranean-style dietary pattern. We selected the aMED diet score as it is consistently associated with lower concentrations of inflammatory and endothelial dysfunction biomarkers, and its reduced prevalence of metabolic syndrome, as characterised by a constellation of metabolic abnormalities, including insulin resistance, dyslipidemia, hypertension, and central obesity—factors that are central to MS progression and symptom burden [40,41]. The aMED allocates points for higher intakes of vegetables, fruits, legumes, whole grains, nuts, fish, a higher monounsaturated-to-saturated fat ratio, moderate alcohol consumption, and lower intakes of red and processed meats. Total aMED scores range from 0 to 9, with higher values indicating greater alignment to a Mediterranean diet.

The MIND diet score is derived from 15 dietary components, including 10 brain-healthy food groups (green leafy vegetables and other vegetables, berries, beans, nuts, whole grains, poultry, fish, olive oil, and wine) and 5 food groups classified as unhealthy to be limited (red meats, butter and margarine, cheese, pastries and sweets, and fried or fast food) [28]. Briefly, each dietary component is scored according to intake frequency, with total scores ranging from 0 to 15, with higher values indicating greater alignment with the MIND dietary pattern. Full component definitions and scoring for both indices are provided in Appendix A.

For analyses, aMED and MIND diet scores were modelled both as continuous variables and as categorical tertiles representing low, moderate, and high alignment with each dietary pattern, consistent with prior work [28,32]. This approach allowed us to evaluate differences in cognitive performance across varying adherence levels and to identify potential cut-points that may inform future dietary interventions or clinical guidance. Total energy intake was estimated from FFQ data (kcal/day). As with all self-reported dietary data, FFQs are subject to misreporting [42]; however, although the FFQ could not be directly validated in our study, we restricted analyses to participants with plausible values (800–4000 kcal/day for males; 500–3500 kcal/day for females) and adjusted total energy intake in models to improve data validity [41].

### 2.4. Cognitive Outcome Measure

Cognitive performance was assessed using the C-MS battery of the Cognitron platform [36]. The C-MS is an MS-optimised, online cognitive battery comprising 13 selected gamified, self-administered tasks targeting domains commonly affected in MS, including memory, attention, executive function, verbal and spatial reasoning, and information processing speed. The complete list of the 13 cognitive tasks across four domains (objects memory, problem solving, information speed and words memory) are outlined in Table 1; Objects Memory Immediate, Objects Memory Delayed, 2D Manipulations, Blocks, Card Pairs, Switching Stroop, Verbal Analogies, Word Definitions, SRT (Simple reaction time), Motor Control, Trail Making, Words Memory Immediate and Delayed. UKMSR participants completed tasks independently on personal devices (smartphones, tablets, or computers).

Normative data from the Great British Intelligence Test (GBIT), which included healthy controls aged 16–90 years with no history of neurological or psychiatric conditions and no prior task exposure, were used as the reference for C-MS scoring. Individual cognition scores were adjusted using regression-based norms to account for sociodemographic and device differences, then standardised to the control-group standard deviation (SD). Cognitive outcomes are expressed as Deviation-from-Expected (DfE) values, representing the number of SDs above or below the expected score for a person with similar characteristics using a similar device.

Primary cognition task outcomes comprised composite indices summarising global and domain-specific performance and assessed both accuracy and latency aspects of cognitive performance. Composites indices were derived using factor analysis of primary scores across tasks: the first unrotated component was taken as “global” cognition (global G factor) to represent overall cognition, and four rotated components (with eigenvalues > 1, per Kaiser’s criterion) of “domain-specific performance” were identified as object memory, problem Solving, information processing speed (IPS), and words memory (Appendix A). For latency outcomes (IPS domain and tasks of Simple Reaction Time, Motor Control and Trail Making), lower DfE values indicate better (faster) performance; whereas for accuracy outcomes, higher DfE values indicate better performance (Table 1).

### 2.5. Statistical Analysis

#### 2.5.1. Primary Analyses

Analysis of participant data with complete observations (complete case analysis) was performed. We assessed both linear associations between diet (with aMED and MIND scores assessed as continuous variables) and cognitive function and non-linear associations between diet (with aMED and MIND scores assessed as categorical tertiles) and cognitive function as in prior dietary studies [43,44]. Linearity and homoskedasticity were assessed using residual plots and the Breusch-Pagan test [45], and showed no strong evidence of heteroskedasticity, supporting the use of linear regression for analyses. Normality and skewness were evaluated using the Shapiro–Wilk and skewness/kurtosis tests.

Multiple linear regression models were used, with robust standard errors, to address residual heteroskedasticity. Estimates are reported as beta coefficients (β) with 95% confidence intervals (CIs). Continuous diet scores were standardised (mean = 0, SD = 1) to facilitate comparability of effect sizes across aMED and MIND diet scores, and DfE scores were used as the primary cognitive outcomes. For linear models, β coefficients represent the expected change in cognitive performance (in SD units) per 1 SD increase in diet alignment. These were interpreted using Cohen’s thresholds for small (β = 0.2), moderate (β = 0.5), and large (β = 0.8) effects. For non-linear models using diet score tertiles, regression coefficients (β) reflect absolute differences in DfE cognitive scores relative to the reference group and were interpreted directly in SD units. Sociodemographic differences including age, sex, dominant hand (left- or right-handed), ethnicity, first language, country of residence, education, occupation, and device being used were accounted for in the outcome standardisation. Regression models were further adjusted for total energy intake, MS phenotype, disease duration, and current use of DMTs.

#### 2.5.2. Subgroup Analyses

Subgroup analyses were conducted to assess differences in associations between diet and cognitive performance between groups defined by MS phenotype (“relapsing” MS, categorised to include benign, RRMS, or those transitioning from RRMS to SPMS) vs. “progressive” MS (SPMS or PPMS), and current DMT usage (yes vs. no). Of note, effective DMTs for progressive MS were limited at the time of data collection, therefore DMT-stratified analyses were restricted to relapsing MS. Intergroup differences and significances thereof were assessed through multiplicative interaction, estimating a product term between the diet and dichotomous interaction terms, the significance of which denoted that of the intergroup difference.

#### 2.5.3. Sensitivity Analyses

A series of sensitivity analyses were conducted to test the robustness of primary analyses examining associations between diet and cognitive function. Firstly, as cognitive tasks were designed to assess distinct domains with minimal inter-correlation, Bonferroni correction was applied within each cognitive domain. This correction was performed by adjusting the significance threshold based on the number of tasks per domain to control the family-wise error rate as sensitivity analysis [46]. For the four domain composites, the significance threshold was adjusted for the four domains (object memory, problem-solving, IPS, and word memory; 0.05/4 = 0.0125). For individual tasks, thresholds were adjusted within each domain according to the number of tasks in the respective domain. This approach controlled the family-wise error rate while respecting the distinct structure of the C-MS battery. Secondly, a rank-based inverse normal transformation was applied to assess the robustness of findings and mitigate the influence of non-normal distributions (Appendix A). Thirdly, Winsorisation was applied to remove outliers, defined as values beyond the 1st and 99th percentiles. These were capped at respective percentile thresholds to ensure that extreme values did not unduly influence any associations observed in primary analyses (Appendix A).

Analyses were conducted using Stata/SE version 18.0 (StataCorp LLC, College Station, TX, USA). Appendix A summarises the study design and analytical workflow, including participant selection, data linkage, and the primary and secondary analyses.

## 3. Results

### 3.1. Participant Characteristics

The study cohort comprised 967 people living with MS enrolled in the UKMSR with complete data, after excluding data from 26 participants with implausible energy intake (Table 2). The mean age was 56.5 (±10.9) years, and average disease duration was 18.8 (±11.7) years. Participants were predominantly female (79.2%), identified as White (97.4%), resided in the UK (99.6%), and nearly half held a university degree (48.9%). The mean aMED and MIND diet scores were 4.2/9 and 7.9/15, respectively.

### 3.2. Characteristics of Cognitive Performance Scores

Cognitive performance was evaluated using mean DfE scores, expressed as standardised values relative to expected performance after accounting for sociodemographic factors and device type, as described in the Methods. Cognitive performance was compared across MS phenotypes (relapsing vs. progressive MS) and by current DMT use (Figure 1). Participants with progressive MS had worse cognitive performance than those with relapsing MS, consistent with the literature [7]. In particular, participants with progressive MS had significantly higher DfE scores on IPS tasks of SRT, Motor Control, and Trail Making (indicating poorer performance), all with *p* < 0.01. Participants with progressive MS also performed significantly worse on 2D Manipulations (*p* < 0.01) and Word Definition (*p* < 0.05). In participants with relapsing MS, those currently taking DMTs also had higher DfE scores compared with those not taking DMTs, although these differences did not reach statistical significance.

### 3.3. Overall Associations Between Mediterranean Diet Adherence and Cognitive Performance

Higher aMED scores were modestly associated with better performance in the IPS (PC2) domain and selected verbal tasks (Table 3). For each 1-SD higher aMED score, IPS DfE was 0.07 SD lower (β = −0.07, 95% CI −0.13 to −0.01) with negative values indicating faster processing speed relative to healthy controls with similar sociodemographic characteristics. This association appeared to be driven by SRT (β = −0.13, 95% CI −0.25, −0.00) and Motor Control (β = −0.12, 95% CI −0.23, −0.02). Higher aMED was also associated with better performance on Verbal Analogies (β = 0.07, 95% CI 0.01 to 0.13) and Word Definitions (β = 0.08, 95% CI 0.01 to 0.15), reflecting better verbal reasoning and crystallised intelligence.

Larger effect sizes were observed when assessing non-linear relationships using categorical aMED (Table 3). Compared with the 1st/lowest aMED tertile, 2nd/middle tertile aMED showed better IPS (0.20 SD lower IPS, 95%CI −0.33, −0.07) and better Trail Making (0.39 SD lower Trail Making, 95% CI −0.72, −0.06). For accuracy-based verbal tasks, the middle aMED tertile scored 0.25 SD higher on Verbal Analogies (95% CI 0.11, 0.39, *p* < 0.001), and the 3rd/highest aMED tertile scored 0.19 SD higher on Word Definitions (95% CI 0.03, 0.36).

Associations between higher aMED and the IPS domain and Verbal Analogies remained significant after within-domain multiple-comparisons adjustment (Table 3). These results were consistent when models were re-estimated using rank-based inverse-normal transformation and Winsorisation of the cognitive outcomes, with significant associations observed between higher aMED and IPS and tasks of Verbal Analogies, Word Definitions, SRT, and Motor Control (Appendix A).

### 3.4. Overall Associations Between MIND Diet Adherence and Cognitive Performance

Higher MIND scores were associated with better performance in IPS and words memory domains, as well as within the IPS domain task of SRT, the problem-solving domain task of Word Definitions and the words memory domain tasks of Words Memory Immediate and Words Memory Delays (Table 4). In particular, categorical MIND was significantly associated with better IPS (2nd tertile: *β* = −0.15, 95% CI −0.28, −0.01) but no significant linear association was found between continuous MIND and IPS. This association was likely driven by the IPS domain task of SRT, with associations found in both continuous and categorical models. Similar inverse trends were observed in IPS domain tasks of Motor Control (Total: *p* = 0.084; 2nd tertile: *p* = 0.065) and Trail Making (2nd tertile: *p* = 0.088).

Higher MIND scores were also associated with better performance in the words memory domain (*β =* 0.06, 95% CI 0.01, 0.12), with a similar pattern observed for moderate MIND adherence (2nd tertile: *β* = 0.17, 95% CI 0.02, 0.31). Within the words memory domain tasks, both Words Memory Immediate (*p* = 0.074) and Words Memory Delayed (*p* = 0.070) showed trending associated with the total MIND. In categorical models, both tasks were significantly associated with moderate MIND diet adherence.

In sensitivity analyses, associations with SRT and Word Definitions (Table 4) remained significant after within-domain multiple-comparisons adjustment. These findings were robust to alternative model specifications, including inverse-normal transformation and Winsorisation. This indicated significant associations were consistently observed between MIND and the cognitive domains of IPS and words memory, and individual tasks of SRT, Words Memory Delayed, and Words Memory Immediate, as well as Word Definitions within the problem-solving domain (Appendix A).

### 3.5. Subgroup Analysis of Associations Between aMED Diet Scores and Cognitive Function by MS Phenotype and DMT Usage

We explored whether the associations between aMED and cognitive performance differed by MS phenotype (relapsing vs. progressive MS). For continuous aMED, no significant differences in DfE scores were observed between relapsing vs. progressive MS, after adjusting for disease duration and DMT use (Appendix A). For categorical aMED, stronger associations were observed in progressive MS, particularly for Trail Making (*p* for interaction = 0.036) and Words Memory Immediate (*p* for interaction = 0.006). In relapsing MS, associations between aMED and cognitive performance were not reliably modified by DMT use (Appendix A).

### 3.6. Subgroup Analysis of Associations Between MIND Diet Scores and Cognitive Function by MS Phenotype and DMT Usage

When assessing MIND-cognition by MS phenotype (relapsing vs. progressive MS), associations between MIND and DfE scores were generally stronger and more consistent in progressive MS than relapsing MS (Appendix A). Among participants with progressive MS, higher MIND scores were associated with better global cognition (G factor), better performance on IPS and words memory domains, and better performance on tasks of Motor Control (IPS domain), Word Definitions (problem solving domain), and the Word Memory Delays and Immediate (words memory domain). In contrast, no such associations were observed in relapsing MS. However, evidence of effect modification by MS phenotype was limited: a significant interaction was observed only for Word Memory Immediate when MIND was modelled categorically, with participants in the moderate tertile showing stronger associations in progressive MS (*p* for interaction = 0.041).

However, in relapsing MS, MIND-cognition associations showed consistent effect modification by current DMT use, after controlling for MS phenotype and disease duration (Figure 2 and Appendix A). Higher MIND scores were associated with better cognitive performance among those not currently taking DMTs versus those taking DMTs, with significant group differences observed for the G factor, IPS, and tasks of 2D Manipulations, Blocks, Motor Control, SRT, Object Memory (Delayed), Trail Making, Word Definitions, and Word Memory (Immediate).

## 4. Discussion

This cross-sectional study found greater alignment with Mediterranean and MIND dietary patterns (i.e., higher aMED and MIND scores) were associated with better cognitive scores, particularly in IPS and words memory domains, and Verbal Analogies and Word Definitions tasks of the problem-solving domain. Whilst effect sizes were modest, these associations were generally consistent in both primary analyses and sensitivity analyses, supporting an overall pattern. In subgroup analyses, associations for both Mediterranean and MIND diets were stronger in people with progressive MS than in those with relapsing MS. Moreover, in those with relapsing MS, associations between MIND and cognitive scores were stronger in people not taking DMTs than in those taking DMTs. Overall, these results suggest that greater alignment with Mediterranean and MIND diets may be associated with better cognitive performance in people living with MS.

### 4.1. Key Findings

We measured cognitive function using the MS-specific C-MS battery which consists of 13 tasks categorised into four domains to characterise broad and disease-sensitive cognitive profiles specific for people living with MS [36]. Overall, our findings indicate that cognitive impairment in people living with MS may be domain-specific, consistent with other studies in the literature [17,22,36]. Further, associations with global cognitive function representing general intelligence (G factor) were only found in subgroup analyses. Whilst this may occur because the G factor explains less than 30% of the variance in the C-MS battery, meaning the G factor may not be able to capture the full breadth of cognitive functions, our results illustrate the value of assessing multiple cognitive function domains, particularly those frequently affected in MS such as processing speed and memory [36,47], rather than relying on a single global score.

The observed associations were modest, which is expected given the observational design of this study and the use of dietary scores reflecting general alignment with dietary patterns rather than adherence to a prescribed intervention. Nevertheless, these findings are consistent with the expectation that diet exerts small but cumulative influences on cognition in complex conditions such as MS. Because cognitive decline arises from multiple interacting inflammatory and neurodegenerative processes, diet alone is unlikely to produce large independent effects. However, even modest associations may have meaningful implications at the population level, particularly when sustained over time or combined with other healthy behaviours.

When specifically examining Mediterranean diet scores, higher aMED was associated with better performance on IPS (a domain reflecting visuomotor processing and mental speed), consistent with the cross-sectional study by Katz Sand et al. [22]. In this study (*N* = 563 people living with MS), greater Mediterranean diet adherence, as assessed by the Mediterranean Diet Adherence Screen (MEDAS), was associated with better performance in the Symbol Digit Modalities Test (SDMT) of the MS Functional Composite test, indicating better cognitive processing speed. Similarly, a small single arm pilot study found greater Mediterranean diet adherence (*N* = 15 people living with MS) resulted in significantly improved cognition at 12 weeks in a range of cognitive tests; SDMT (*p* = 0.006; d = 1.22), the cognitive component of the Timed Up and Go (TUG) test (*p* = 0.01; d = 0.93) and the MSQOL-29 cognitive subscale (*p* = 0.03; d = 0.61) [17]. Higher aMED was also associated with greater verbal ability (higher performance on Word Definitions and Verbal Analogies), similar to the findings of a recent systematic review where greater Mediterranean-style diet adherence in cross-sectional studies was associated with better verbal ability in people with neurodegenerative diseases (Alzheimer’s disease and dementia) [48].

However, RCTs in the literature have found greater Mediterranean diet adherence did not result in better cognitive outcomes in people living with MS. In a small RCT (*N* = 36 people living with MS), adherence to a modified Mediterranean diet did not improve 6-month cognitive outcomes, as assessed by the Minimal Assessment of Cognitive Function in Multiple Sclerosis (MACFIMS) battery [20]. Likewise, an RCT of 147 people living with MS from Iran found adherence to a modified Mediterranean diet vs. a traditional Iranian diet did not result in improved cognitive function at 6 months, as measured by the cognitive component of the MSQOL-54 [25]. However, statistically significant improved cognitive function (as assessed by a 5.6-point reduction (*p* = 0.027) in the cognitive subscale of the Modified Fatigue Impact Scale (MFIS)) was observed.

Conflicting Mediterranean diet–cognition study findings may reflect the complexity of this relationship, which is further influenced by differences in study design, sample sizes and characteristics, measures of dietary intake, cognitive function tests employed and external factors such as adherence to other lifestyles such as diet, physical activity and omega-3 [16,17,18,19]. Challenges related to study feasibility, dietary adherence and how the diet is defined and implemented [49], may further contribute to weaker or null effects in diet-related RCTs involving people living with MS.

When examining MIND diet-cognition relationships, higher MIND scores were associated with better scores on the IPS domain, specifically the SRT, as well as the two tasks of the words memory domain (Words Memory Intermediate and Words Memory Delays). Although no comparable studies have examined MIND diet scores in people living with MS, our findings align with the findings of a longitudinal study of older adults with a history of stroke where greater MIND diet adherence, but not Mediterranean or DASH diet adherence, was associated with slower rates of decline in global cognition, semantic memory, and perceptual speed [32]. Moreover, a recent RCT involving middle-aged, obese women found improvements in cognition after 3 months adherence to a MIND diet, with longer-term adherence linked to better verbal memory performance [50,51].

### 4.2. Group Differences

Our analyses indicated that associations between MIND scores and cognitive function may differ by MS phenotypes and DMT use. That is, MIND-cognition associations were stronger in those with progressive vs. relapsing MS and in those with relapsing MS currently not taking DMTs vs. those not taking DMTs. In particular, amongst relapsing MS cases not currently taking DMTs, higher MIND diet scores were associated with better performances in global cognition, IPS and tasks of Words memory in participants, whereas little to no association was seen in those currently taking DMTs, with significant group differences. However, given the crude dichotomisation of DMT use and the cross-sectional study design, this observation should be interpreted with caution. The pattern also likely reflects bias and confounding. That is, the cohort of people not taking DMTs may have different characteristics to those taking DMTs, which may explain why they have better cognitive function (i.e., they may be healthier and/or engage in healthier lifestyles or other tasks or self-management strategies which are beneficial for their cognitive function) [18,52,53,54]. Nevertheless, these findings suggest future longitudinal and/or interventional studies to clarify the contribution of diet to cognitive outcomes across treatment contexts are warranted.

In terms of positive cross-sectional associations observed between Mediterranean and MIND diet scores and better cognition, this may reflect the emphasis of these diets on foods that support brain health. That is, leafy green vegetables and berries—key components of the MIND pattern—are rich in antioxidants and bioactive compounds that are independently associated with slower cognitive decline and improved memory [30,55], and may explain the positive memory-related associations we observed. Further, a longitudinal study of adults aged ≥65 years followed for 10 years found higher intake of flavonoid, found in plants such as vegetables and fruits, was associated with less decline in cognition function [56], which may explain why the plant-rich diets offer cognitive benefits [20].

Biologically plausible explanations for observed Mediterranean and MIND diet-cognition associations could be explained by anti-inflammatory and oxidative-stress pathways, resulting in neuroprotective mechanisms. In addition, reduced oxidative stress supports vascular health, and subsequently brain health. In particular, these pathways could complement pharmacological suppression and may be beneficial in the absence of pharmacological suppression, especially for processing speed, a domain sensitive to diffuse network integrity and vascular comorbidity [57]. These anti-inflammatory and antioxidant effects may also act through the gut–brain axis [20,58]. Diet also influences the composition and metabolic activity of the gut microbiota, which regulates immune function and generates metabolites such as short-chain fatty acids, polyphenols, and tryptophan derivatives that can cross the blood–brain barrier and modulate neuroinflammatory processes. Both Mediterranean and MIND diets emphasise fibre- and polyphenol-rich plant foods that support a beneficial gut microbial profile, providing a plausible mechanistic pathway linking diet quality with cognitive resilience in MS.

Evidence for effect modification of MS phenotype was less robust. Associations between diet scores and cognitive performance in IPS and words memory domains, as well as Verbal Ability were generally larger and more consistent in progressive MS, than in relapsing MS. However, formal tests of interaction by phenotype were only significant for two tasks—Trail Making (aMED only) and Word Memory Immediate (both aMED and MIND)—and no domain level interaction reached significance. While direct research on diet–cognition links stratified by MS phenotype is lacking, our results support evidence in the literature that diet may differentially influence outcomes in progressive MS. For instance, Nag et al. analysed data from 1108 people with MS in the iConquer MS registry (predominantly US-based) and found associations between anti-inflammatory diets, including the Mediterranean diet, and better mobility outcomes were stronger in people with progressive MS than in those with non-progressive MS [59]. Similarly, Katz Sand et al. reported stronger protective effects of Mediterranean diet adherence on disability composite scores, including cognition composite scores, amongst individuals with progressive MS than in those with relapsing MS [22]. Taken together, these findings are compatible with the hypothesis that healthier dietary patterns may support brain health which may be beneficial for cognitive function in people with progressive MS.

The Mediterranean and MIND dietary patterns have been positively associated with cognitive outcomes in non-MS populations, including healthy older adults and those with neurodegenerative diseases [18,19,23,24]. The present findings from our MS cohort align with these broader associations and extend the evidence to people with MS, in whom disease-specific mechanisms such as neuroinflammation, oxidative stress, and demyelination may further influence these relationships. Further studies investigating these dynamics, including imaging-based assessments in clinical and laboratory settings, are needed to better understand these pathways and to clarify how these diets may affect cognition.

### 4.3. Strength and Limitations

Key strengths of this study include the large population-based UKMSR dataset whose robustness and validity has been demonstrated through several clinically relevant studies and high-impact publications [36,60,61]. Furthermore, cognitive impairment was measured using the validated, MS-specific C-MS cognitive battery across four cognitive domains relevant for people living with MS, and the use of DfE scores adjusted for sociodemographic factors both helped to minimise confounding and allowed for reliable and fair comparisons across individuals. The large normative dataset from the GBIT (~66,000 participants) provided a robust reference for these comparisons [36], and comprehensive dietary intake data was captured via the detailed 130-item FFQ, which has been previously used in people living with MS [60], and validated in large-scale nutrition research [62].

Comprehensive analyses, including both sensitivity and subgroup analyses, were performed. To provide a comprehensive picture of diet-cognition associations, we assessed both linear and non-linear relationships. Notably, categorical models are particularly informative as they capture behavioural distinctions between groups and are less affected by regression dilution. In these analyses, participants in the middle and highest dietary tertiles performed up to approximately 0.4 SD better on certain cognitive measures than those in the lowest tertile, with associations for some domains appearing stronger in the middle tertile than the highest. This pattern is consistent with threshold effects reported in other nutrition studies [63,64], and suggests that cognitive benefits may become apparent once a certain level of dietary alignment is reached (e.g., moving from low to moderate adherence), with further increases in alignment only yielding smaller gains.

However, several limitations should be considered when interpreting our study findings. First, the cross-sectional design precludes causal inference. It is possible that individuals with better cognitive function were more likely to adhere to healthier diets, rather than dietary adherence influencing cognitive performance. However, the observed diet-cognition associations of our study are still highly informative and lay the groundwork for future studies to examine potential causal pathways. Second, residual confounding from other lifestyle behaviours such as physical activity, smoking status, sleep, and social engagement are likely to exist that were not accounted for in analyses [16,17,18,19]. Given alignment with the Mediterranean and/or MIND diet likely reflects broader health-conscious behaviours, it is challenging to isolate independent effects of diet adherence; however, identifying the contribution of diet from other lifestyle components was beyond the scope of the present study. Third, our diet data obtained via the FFQ was self-reported, thereby potentially introducing recall error, over reporting and social desirability bias. While the FFQ has been used in other studies [60], it is not validated for the MS population. However, we attempted to mitigate this by excluding implausible energy intakes and adjusting nutrient intakes for total energy, although these steps cannot eliminate reporting bias entirely.

MS phenotype and DMT use were also self-reported by participants so misclassification is possible, and future studies linked to data from clinical records would strengthen these studies. Relatedly, more objective dietary assessment methods (e.g., repeated 24 h recalls or biomarkers) could be employed in future research by validating self-reported intake, although their feasibility in large-scale cohorts may be limited by cost and participant burden. In addition, while we discussed potential biological pathways—such as anti-inflammatory, antioxidant, and gut–brain axis mechanisms—these remain speculative within the context of our cross-sectional design. No biomarkers, microbiome, or inflammatory measures were collected, and therefore the hypothesised mechanisms cannot be directly verified in this study. Furthermore, given the modest effect sizes observed, future longitudinal or mechanistic studies incorporating biological markers are needed to clarify causality and to further elucidate the pathways linking dietary patterns with cognitive outcomes in MS.

There is also potential selection bias in our sample. Participants who completed both the extensive diet questionnaire and the lengthy cognitive testing may represent a healthier and/or more motivated subset of the MS population. However, evidence from the UKMSR C-MS study suggests that attrition may not disproportionately exclude the most impaired individuals; as those returning for follow-up were on average older, had longer disease duration, and had characteristics typically associated with greater cognitive impairment [36]. Furthermore, the demographic profile of the sample—predominantly white, highly educated individuals residing in the UK may limit the generalisability of our findings to more diverse populations or those in other healthcare settings. Lastly, while we pre-specified our primary domain-level outcomes, the large number of tasks conducted across tasks may increase the risk of false-positive findings as some associations did not remain statistically significant after correcting multiple comparisons. Nonetheless, consistent directional effects, domain-level clustering, and robustness in sensitivity analyses help support the validity of the diet-cognition patterns we observed.

## 5. Conclusions

Cognitive impairment is a common feature of MS, yet its management remains challenging [10]. In this context, our cross-sectional analyses indicated greater alignment with Mediterranean and MIND dietary patterns were associated with better cognitive function in IPS, words memory, and Verbal ability—areas of cognitive impairment frequently affected in MS [65]. Notably, associations between MIND diet scores and cognitive performance also differed by DMTs use amongst participants with relapsing MS. Whilst our results provide insights into the potential clinical relevance of dietary patterns across specific cognitive domains and within MS subgroups (progressive vs. relapsing MS), replication and validation of our study findings in longitudinal and/or interventional studies is needed to inform future dietary recommendations for people living with MS. Such studies should integrate detailed dietary assessment with gut–brain axis measures and inflammatory/immune and redox biomarkers to elucidate mechanisms and test causality.

## Figures and Tables

**Figure 1 nutrients-17-03326-f001:**
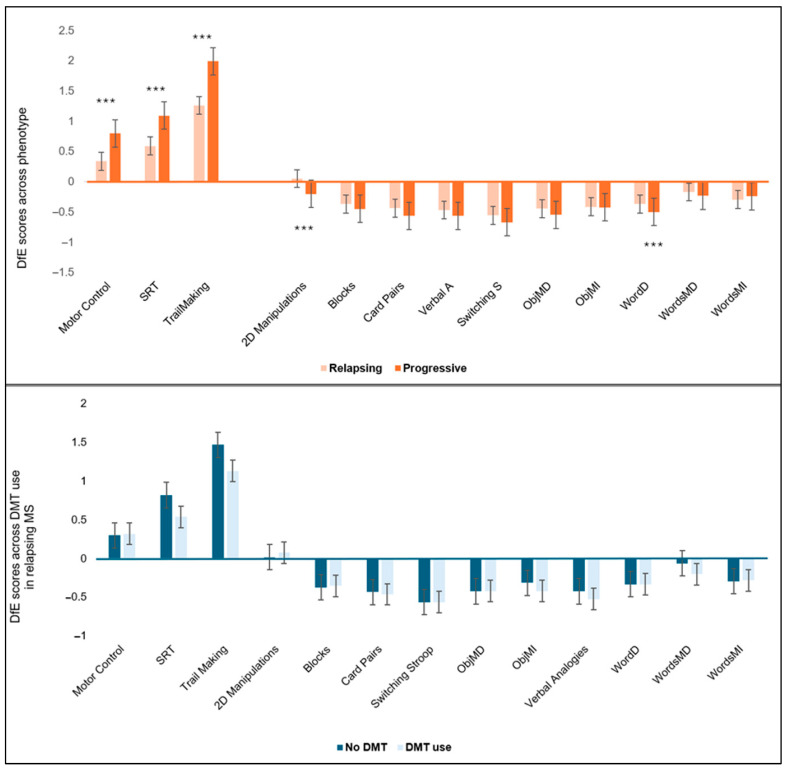
Examination of cognitive performance across phenotype of multiple sclerosis (**above**) and DMT use in relapsing MS (**below**). Note: Cognitive performance was assessed using Deviation from Expected (DfE) values, adjusted for age, sex, handedness, ethnicity, first language, country of residence, education, occupation, and device being used using normative data from the Great Britain Intelligence Test. *T*-test was applied for each task to test whether group differences: *** *p* < 0.001. Abbreviations: DfE = Deviation from Expected (adjusted cognitive scores standardised using the control SD to reflect the number of SDs above or below the expected score for someone with similar sociodemographic characteristics who completed the assessment on a similar device); DMT = Disease-Modifying Therapy; MS = multiple sclerosis; ObjMD = Objects Memory Delayed; ObjMI = Objects Memory Immediate; Switching S = Switching Stroop; SRT = Simple Reaction Time; Verbal A = Verbal Analogies; WordD = Word Definitions; WordsMD = Words Memory Delayed; WordsMI = Words Memory Immediate.

**Figure 2 nutrients-17-03326-f002:**
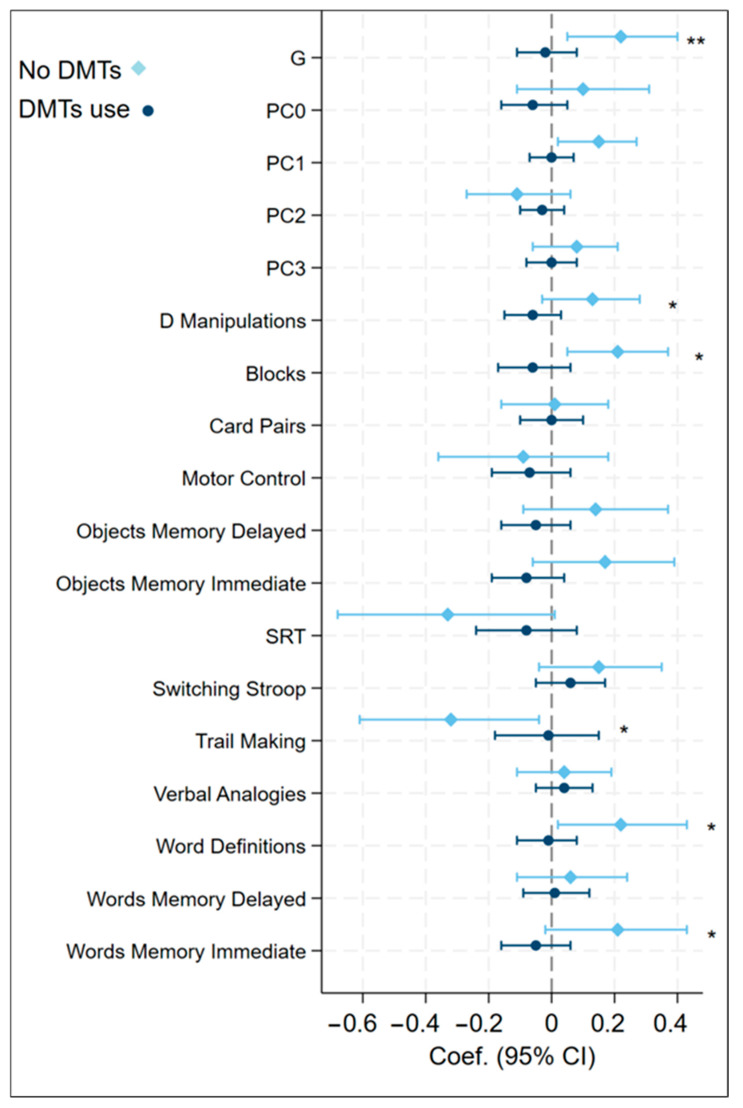
Effect modification in relapsing MS: associations between MIND diet score and DfE scores in DMT use vs. no DMT groups. Note: Significant group difference: * *p* for interaction effect < 0.05; ** *p* for interaction effect <0.01 Abbreviations: DMT = Disease-Modifying Therapy; G = general intelligence factor, MIND = Mediterranean-DASH Intervention for Neurodegenerative Delay; ObjMD = Objects Memory Delayed; ObjMI = Objects Memory Immediate; PC0 = object memory; PC1 = problem solving, PC2 = information processing speed; PC3 = words memory; Switching S = Switching Stroop; SRT = Simple Reaction Time; Verbel A = Verbal Analogies; WordD = Word Definitions; WordsMD = Words Memory Delayed; WordsMI = Words Memory Immediate.

**Table 1 nutrients-17-03326-t001:** Cognitive measures from the C-MS battery.

Cognitive Domains	Cognitive Task	Cognitive Domain Measured	Task Description	Score
G factor	All	General intelligence	Aggregate score representing overall cognitive performance across all tasks.	Accuracy
P0: Objects memory	Objects Memory Immediate	Short-term visual recognition memory	Identify target object from distractors immediately after viewing a series of object silhouettes.	Accuracy
	Objects Memory Delayed	Medium-term visual recognition memory	Recall previously seen objects after a delay without being reminded of them.	Accuracy
P1: Problem solving	2D Manipulations	2D spatial reasoning	Match a rotated grid of coloured squares with the correct unaltered version.	Accuracy
	Blocks	2D spatial planning	Remove blocks to match a target configuration in a gravity-affected grid.	Accuracy
	Card Pairs	Associative working memory	Match pairs of cards after they have been turned over from memory.	Accuracy
	Switching Stroop	Attentional control and cognitive flexibility	Respond based on changing conditions of text or ink colour under rule-switching.	Accuracy
	Verbal Analogies	Grammar-based verbal reasoning	Evaluate the correctness of analogy statements based on grammar and logic.	Accuracy
	Word Definitions	Crystallised verbal knowledge	Choose the correct definition for a series of English words from multiple choices.	Accuracy
P2: Information processing speed	SRT	Simple reaction time	Click on the screen as soon as a visual target appears.	Latency
	Motor Control	Visuomotor abilities	Click on targets that appear across the screen as quickly and accurately as possible.	Latency
	Trail Making	Cognitive flexibility and visual attention	Click on tiles in numerical and alphabetical order, alternating between them.	Latency
P3: Words memory	Words Memory Immediate	Short-term verbal recognition memory	Identify previously seen words from a new list immediately after initial exposure.	Accuracy
	Words Memory Delayed	Medium-term verbal recognition memory	Identify previously seen words from a new list after a delay.	Accuracy

Note: DfE scores are standardised relative to healthy controls (SD units). For accuracy outcomes, higher DfE = better performance; for latency outcomes, higher DfE = worse/slower performance. (1.0 on the DfE scale corresponds to 1 SD).

**Table 2 nutrients-17-03326-t002:** Characteristics of participants, *N* (%).

Demographics		*N* (%)
Total *N*		967
Age (M, SD)		56.47 (10.86)
MS duration (M, SD)		18.8 (11.67)
Gender	Female	766 (79.2)
	Male	201 (20.8)
Ethnicity ^a^	White	942 (97.4)
	Asian or Asian British	8 (0.8)
	Black/African/Caribbean/Black British	5 (0.5)
	Mixed or multiple ethnic groups	8 (0.8)
	Other	4 (0.4)
Dominant hand	Right	831 (85.9)
	Left	105 (10.9)
	Ambidextrous	31 (3.2)
First language	English	937 (96.9)
	Other	30 (3.1)
Residence	United Kingdom	963 (99.6)
	Abroad	4 (0.4)
Education	Pre-General Certificate of Secondary Education	22 (2.3)
	School	439 (45.4)
	Degree	473 (48.9)
	PhD	33 (3.4)
Occupation	Worker	347 (35.9)
	Retired	399 (41.3)
	Disabled/Not applicable/Sheltered employment	163 (16.9)
	Homemaker	41 (4.2)
	Unemployed/Looking for work	12 (1.2)
	Student	3 (0.3)
	Unknown	2 (0.2)
MS phenotype	Benign	30 (3.0)
	RRMS	514 (51.8)
	PPMS	159 (16.0)
	SPMS	233 (23.5)
	Transitioning to SPMS	27 (2.7)
	Unknown	30 (3.0)
DMT use	Yes	401 (51.4)
	No	379 (48.6)
aMED diet score (M, SD)		4.20 (1.99)
MIND diet score (M, SD)		7.85 (1.67)

Note: ^a^ Ethnicity categories reflect the fixed response options in the source questionnaire; more granular national identities (e.g., British, Irish and African/Caribbean subgroups) were not collected. Abbreviation: aMED = alternative Mediterranean; DMT = disease modifying therapy; MIND = Mediterranean-DASH Intervention for Neurodegenerative Delay; M = mean; MS = multiple sclerosis; PhD = Doctor of Philosophy; SD = standard deviation; MS Phenotype was defined as relapsing MS (benign MS, relapsing-remitting MS [RRMS] and transitioning to secondary progressive MS [SPMS]) and progressive MS (SPMS and primary progressive MS [PPMS]).

**Table 3 nutrients-17-03326-t003:** Overall associations between Mediterranean diet (assessed as continuous and categorical tertile aMED) vs. cognitive function scores.

	Total aMED (0–9)			Categorical aMED			
				T2: aMED (4–5)		T3: aMED (6–9)	
Domain	Test	β (95% CIs)	*p*	β (95% CIs)	*p*	β (95% CIs)	*p*
G factor		0.04 [−0.03, 0.10]	0.278	0.03 [−0.13, 0.19]	0.742	0.08 [−0.08, 0.25]	0.315
PC0: Object memory		−0.05 [−0.12, 0.02]	0.174	−0.14 [−0.30, 0.03]	0.103	−0.12 [−0.30, 0.05]	0.169
	Objects In Memory Delayed	−0.04 [−0.12, 0.03]	0.271	−0.13 [−0.31, 0.05]	0.259	−0.06 [−0.26, 0.13]	0.526
	Objects Memory Immediate	−0.04 [−0.12, 0.04]	0.333	−0.13 [−0.31, 0.05]	0.155	−0.11 [−0.31, 0.08]	0.304
PC1: Problem solving		0.02 [−0.04, 0.07]	0.549	−0.03 [−0.17, 0.10]	0.642	0.03 [−0.11, 0.18]	0.678
	2D Manipulations	−0.02 [−0.09, 0.05]	0.566	0.03 [−0.14, 0.19]	0.761	−0.07 [−0.25, 0.10]	0.427
	Blocks	0.04 [−0.05, 0.12]	0.336	−0.15 [−0.34, 0.04]	0.123	−0.06 [−0.27, 0.14]	0.546
	Card Pairs	−0.02 [−0.09, 0.05]	0.549	−0.04 [−0.21, 0.14]	0.676	−0.05 [−0.24, 0.14]	0.599
	Switching Stroop	−0.02 [−0.05, 0.10]	0.535	−0.13 [−0.32, 0.06]	0.180	0.10 [−0.11, 0.30]	0.352
	**Verbal Analogies**	**0.07 [0.01, 0.13]**	**0.025**	**0.25 [0.011, 0.39]**	**<0.001 †**	0.09 [−0.06, 0.25]	0.219
	**Word Definitions**	**0.08 [0.01, 0.15]**	**0.017**	0.04 [−0.13, 0.20]	0.673	**0.19 [0.03, 0.36]**	**0.023**
**PC2: IPS**		**−0.07 [−0.13, −0.01]**	**0.013**	**−0.20 [−0.33, −0.07]**	**0.005 †**	0.13 [−0.02, 0.27]	0.089
	**SRT**	**−0.13 [−0.25, −0.00]**	**0.045**	−0.17 [−0.45, 0.11]	0.243	−0.24 [−0.55, 0.06]	0.112
	**Motor Control**	**−0.12 [−0.23, −0.02]**	**0.020**	−0.20 [−0.44, 0.04]	0.101	−0.22 [−0.45, 0.02]	0.069
	**Trail Making**	−0.12 [−0.25, 0.01]	0.068	**−0.39 [−0.72, −0.06]**	**0.020**	−0.25 [−0.61, 0.12]	0.183
PC3: Words memory		0.04 [−0.02, 0.10]	0.240	−0.03 [−0.18, 0.11]	0.664	0.08 [−0.07, 0.23]	0.304
	Words Memory Immediate	0.06 [−0.03, 0.14]	0.192	0.13 [−0.06, 0.33]	0.179	0.15 [−0.06, 0.36]	0.165
	Words Memory Delays	0.04 [−0.04, 0.12]	0.299	0.10 [−0.09, 0.30]	0.303	0.07 [−0.13, 0.28]	0.495

Note: Cognitive scores are expressed as Deviation from Expected (DfE) values, adjusted for age, sex, handedness, ethnicity, first language, country of residence, education, occupation, and device being used using normative data from the Great British Intelligence Test (GBIT). Models were adjusted for total energy intake, MS phenotype, disease duration and DMT use. Regression coefficients (β) represent the standard deviation (SD) difference in cognitive performance relative to the expected score for individuals with similar sociodemographic characteristics, per 1 SD increase in aMED or relative to the lowest tertile (for categorical aMED). Bold values indicate significant results (*p* < 0.05). Abbreviation: aMED = adjusted Mediterranean diet; IPS = Information processing speed; SRT = Simple Reaction Time. Bonferroni correction was applied separately at each level: across the four domain composites (4 tests) and across individual tasks within each domain (number of tasks per domain). † Significant Bonferroni-corrected *p* = 0.05/4 = 0.0125 for PC2 and *p* = 0.05/6 = 0.008 for Verbal Analogies within PC1 domain).

**Table 4 nutrients-17-03326-t004:** Associations between MIND scores and cognitive performance.

	Total MIND			Categorical MIND			
				T2: MIND (4–5)		T3: MIND (6–9)	
Domain	Subtest	β (95% CIs)	*p*	β (95% CIs)	*p*	β (95% CIs)	*p*
G factor		0.07 [−0.01, 0.13]	0.060	0.09 [−0.08, 0.26]	0.298	0.13 [−0.04, 0.30]	0.145
PC0: Object memory		0.01 [−0.06, 0.08]	0.779	0.01 [−0.16, 0.18]	0.895	−0.03 [−0.21, 0.16]	0.788
	Objects Memory Delayed	0.04 [−0.04, 0.12]	0.376	0.05 [−0.14, 0.24]	0.606	0.03 [−0.17, 0.23]	0.770
	Objects Memory Immediate	0.01 [−0.07, 0.10]	0.748	0.03 [−0.16, 0.21]	0.774	−0.02 [−0.21, 0.18]	0.878
PC1: Problem solving		0.04 [−0.02, 0.09]	0.197	−0.03 [−0.17, 0.10]	0.623	0.08 [−0.06, 0.22]	0.243
	2D Manipulations	0.01 [−0.07, 0.08]	0.939	0.07 [−0.10, 0.24]	0.430	0.04 [−0.13, 0.21]	0.622
	Blocks	0.01 [−0.07, 0.09]	0.726	−0.02 [−0.21, 0.17]	0.831	0.03 [−0.17, 0.24]	0.748
	Card Pairs	0.03 [−0.04, 0.10]	0.428	0.07 [−0.10, 0.24]	0.439	0.04 [−0.13, 0.21]	0.682
	Switching Stroop	0.06 [−0.02, 0.14]	0.121	−0.10 [−0.29, 0.05]	0.303	−0.11 [−0.30, 0.09]	0.273
	Verbal Analogies	0.03 [−0.03, 0.09]	0.354	0.07 [−0.08, 0.21]	0.364	0.01 [−0.14, 0.16]	0.927
	**Word Definitions**	**0.09 [0.02, 0.15]**	**0.009 †**	0.06 [−0.11, 0.23]	0.500	**0.21 [0.04, 0.38]**	**0.015**
PC2: IPS		−0.05 [−0.10, 0.01]	0.104	**−0.15 [−0.28, −0.01]**	**0.032**	−0.13 [−0.27, −0.01]	0.064
	**SRT**	**−0.13 [−0.25, −0.01]**	**0.031**	**−0.31 [−0.59, −0.02]**	**0.029**	**−0.38 [−0.67, −0.10]**	**0.014 †**
	Motor Control	−0.08 [−0.18, 0.01]	0.084	−0.23 [−0.47, 0.00]	0.065	−0.20 [−0.46, 0.06]	0.135
	Trail Making	−0.10 [−0.24, 0.04]	0.164	−0.30 [−0.64, 0.06]	0.088	−0.27 [−0.62, 0.08]	0.134
**PC3: Words memory**		**0.06 [0.01, 0.12]**	**0.037**	**0.17 [0.02, 0.31]**	**0.016**	0.11 [−0.03, 0.26]	0.125
	**Words Memory Immediate**	0.07 [0.00, 0.16]	0.074	**0.20 [0.01, 0.39]**	**0.037**	0.11 [−0.09, 0.31]	0.266
	**Words Memory Delays**	0.07 [0.01, 0.15]	0.070	**0.21 [0.02, 0.41]**	**0.031**	0.15 [−0.04, 0.35]	0.127

Note: Cognitive scores are expressed as Deviation from Expected (DfE) values, adjusted for age, sex, handedness, ethnicity, first language, country of residence, education, occupation, and device being used using normative data from the Great British Intelligence Test (GBIT). Models were adjusted for total energy intake, MS phenotype, disease duration and DMT use. Regression coefficients (β) represent the standard deviation (SD) difference in cognitive performance relative to the expected score for individuals with similar sociodemographic characteristics, per 1 SD increase in MIND or relative to the lowest tertile (for categorical MIND). Bold values indicate significant results (*p* < 0.05). Abbreviation: IPS = Information processing speed; MIND = Mediterranean-DASH Intervention for Neurodegenerative Delay; SRT = Simple Reaction Time. Bonferroni correction was applied separately at each level: across the four domain composites (4 tests) and across individual tasks within each domain (number of tasks per domain). † Significant Bonferroni-corrected *p* = 0.05/4 = 0.0125 for PC2 and *p* = 0.05/6 = 0.008 for Word Definitions within PC1 domain).

## Data Availability

The author signed the data access agreement to have access to the Secure eResearch Platform. Therefore, restrictions apply to the availability of these data. Data was obtained from UK MS register and are available from the authors with the permission of the UK MS register.

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
