# Peer review of "Mediterranean and MIND Dietary Patterns and Cognitive Performance in Multiple Sclerosis: A Cross-Sectional Analysis of the UK Multiple Sclerosis Register"

_nutrients, 2025, doi:10.3390/nu17213326_

Round 1
Reviewer 1 Report
Comments and Suggestions for Authors
The article written by Yu and coauthors, entitled :"Mediterranean and MIND dietary patterns and cognitive performance in multiple sclerosis: a cross-sectional analysis of the UK Multiple Sclerosis Register", ID nutrients-3896917, deals with an emerging topic in neurological diseases, even those attaining cognitive degeneration: the therapeutic support of diet and specific nutritional patterns. It also appraise a large cohort of subjects. At the same time, multiple sclerosis is a disease with specific symptoms and progression that are not always directly linked to cognitive impairment. The article is well written and provides new data in a field that has a huge impact on public health, multiple sclerosis, given the potentially progression towards highly debilitating losses of motor skills and, although less frequently considered, cognitive skills in some cases.
The authors should address however some parts and aspects of their manuscript.
iIn the Introduction: a more medical and precise description of multiple sclerosis should be presented, also treating its etiopathogenesis at the molecular and immune level. This is important to relate nutrition to autoimmune features and inflammatory networks specifically altered. Cognitive signs are not always present. An epidemiological prevalence of the illness and of specific cluster of symptoms should be provided too. This will enable authors to refine their aims.
Methods: methods are quite complex. The authors should explain in a clearer way their methodology and study design using for instance schemas and a Figure. The diet intervention should be better explained in order to permit to all readers to easily get information about this aspect. What about anti-inflammatory diets ? Patients should be stratified also in respect to their clinical history in respect to the disease progression. There are some subjects' features that are considered without explanation, such as whether they are right-handed or ambidextrous, and others. The statistical methods should be clearer too. Biases and confounding factors should be addressed.
The discussion should better explain their results. The authors state that they obtained modest results: they should develop this concept on the bases of their significant results. The role of the gut-brain axis should be addressed too to consider possible mechanisms involving nutrients and brain function. Anti-inflammatory features of diet interventions should be specifically treated.
Limitations should be modified considering all these suggestions.
Reviewer 2 Report
Comments and Suggestions for Authors
Congrats on the article, it is really well-done.
This is a cross-sectional study analyzing the associations between diet and cognition data from 967 participants in the United 34 Kingdom Multiple Sclerosis Register.
I just suggest mentioning in the figures the meaning of the abbreviations (DMT, for example).
Author Response
|
Comments 1: Congrats on the article, it is really well-done. This is a cross-sectional study analyzing the associations between diet and cognition data from 967 participants in the United 34 Kingdom Multiple Sclerosis Register. I just suggest mentioning in the figures the meaning of the abbreviations (DMT, for example).
|
|
Response 1: Thank you for your positive feedback. We have added explanations for all abbreviations used in the figures. Change made: Figure 1 notes (line 306): “DMT=Disease-Modifying Therapy” Table 3 notes (line 399)” “IPS=Information processing speed” Table 4 notes (within the table)” “IPS=Information processing speed” Figure 2 note (line 409): “DMT=Disease-Modifying Therapy” |
Reviewer 3 Report
Comments and Suggestions for Authors
The manuscript presents a well-designed cross-sectional analysis of Mediterranean and MIND dietary patterns in relation to cognitive performance among people with multiple sclerosis. The large cohort and use of an MS-specific cognitive battery represent notable strengths.
However, I have some suggestions:
In order to avoid causal inference, some statements could be further moderated while the discussion acknowledges the cross-sectional design. For example, statements claiming that the diet "improves" the mind should be rephrased to emphasize the association rather than the cause.
Among lifestyle factors, it would be useful to know the smoking habits of the study participants and whether this was eventually considered among the confounding factors.
Author Response
|
Comments 1: The manuscript presents a well-designed cross-sectional analysis of Mediterranean and MIND dietary patterns in relation to cognitive performance among people with multiple sclerosis. The large cohort and use of an MS-specific cognitive battery represent notable strengths. However, I have some suggestions:
In order to avoid causal inference, some statements could be further moderated while the discussion acknowledges the cross-sectional design. For example, statements claiming that the diet "improves" the mind should be rephrased to emphasize the association rather than the cause. |
|
Response 1: Thank you for this thoughtful comment. We have carefully reviewed the manuscript to ensure causal language was used appropriately. Terms such as “improves” were only used when referring to findings from other studies (prospective or randomised controlled trials), where causal interpretation is methodologically justified. One change made in the discussion: “4.2 Group differences: added “cross-sectional” in line 497 to emphasise the associations’ nature]”
|
|
Comments 2: Among lifestyle factors, it would be useful to know the smoking habits of the study participants and whether this was eventually considered among the confounding factors. |
|
Response 2: Thank you for this helpful comment. We agree that smoking is an important lifestyle factor that may influence both diet and cognitive outcomes. In this cohort, dietary data were collected at two time points (2016 and 2022), and cognitive data were linked at the 2022 wave. Unfortunately, smoking status was not available at this same timepoint, and using earlier smoking data could introduce misclassification and reduce validity. Therefore, smoking was not included in the current analyses.
We have acknowledged this as a limitation and added the following text to the Discussion: Section 4.3 Strength and limitations (Line 566): “Second, residual confounding from other lifestyle behaviours such as physical activity, smoking status, sleep, and social engagement are likely to exist that were not accounted for in analyses[12-15].
|
Reviewer 4 Report
Comments and Suggestions for Authors
No real comments to make. In my opinion the article is interesting and well written, the research is well documented and well commented as to support the findings without exagerrating their meaning/weight. The number of subjects is large, methods are adequately described and commented, statistical analysis is more than convincing and although the paper isn't ground breaking in its theme, it adds valuable and good quality information that may help practicians and patients change their habits.
However, it is not clear to me whether the associations between the two dietary patterns and cognition in MS patients are specific to MS or are similar to the cognitive changes expected in the general population (or to other particular neurodegenerative diseases) in a similar setting. In other words, is there a supplementary potential impact/benefit of a healthy diet for MS patients, or it is the same as in relatively normal (let's say overweight) subjects? Although I think that the paper is good, I would appreciate if the Authors could comment a bit more on this topic.
Author Response
|
Comments 1: No real comments to make. In my opinion the article is interesting and well written, the research is well documented and well commented as to support the findings without exagerrating their meaning/weight. The number of subjects is large, methods are adequately described and commented, statistical analysis is more than convincing and although the paper isn't ground breaking in its theme, it adds valuable and good quality information that may help practicians and patients change their habits. However, it is not clear to me whether the associations between the two dietary patterns and cognition in MS patients are specific to MS or are similar to the cognitive changes expected in the general population (or to other particular neurodegenerative diseases) in a similar setting. In other words, is there a supplementary potential impact/benefit of a healthy diet for MS patients, or it is the same as in relatively normal (let's say overweight) subjects? Although I think that the paper is good, I would appreciate if the Authors could comment a bit more on this topic. |
|
Response 1: Thank you for this insightful question. The Mediterranean and MIND dietary patterns are not specific to MS and have been associated with better cognitive outcomes in the general population, including healthy older adults and individuals with neurodegenerative conditions such as Alzheimer’s disease and dementia. There have not been any such studies of these diets and cognitive outcomes in people with MS, however. Our study addresses this gap in evidence and for the first time investigates how these diets affect cognitive outcomes in people with MS. The associations we observed between diet and cognition in our MS sample are consistent with findings in non-MS populations. Importantly, however, the pathophysiology of cognitive change in MS includes MS-specific mechanisms(1), on top of those seen in the general population(2). These results could indicate that Mediterranean and MIND diets may be associated with diverse mechanisms underlying cognitive change, both general and MS-specific. However, as we have no ability to discriminate between these pathophysiological aspects, this is mere conjecture at this stage. At the same time, we note that MS-specific factors, including neuroimmunological involvement and associated neuroinflammation, need be considered in how diet may affect cognitive outcomes in MS. Further studies investigating these dynamics, including imaging-based assessments in clinical samples and in laboratory settings, are needed to better understand these pathways and to interpret how these diets may affect cognition.
We added a paragraph in the Discussion to clarify this point:
4.2 Group difference (line 531): “The Mediterranean and MIND dietary patterns have been positively associated with cognitive outcomes in non-MS populations, including healthy older adults and those with neurodegenerative diseases [18,19,23,24]. The present findings from our MS cohort align with these broader associations and extend the evidence to people with MS, in whom disease-specific mechanisms such as neuroinflammation, oxidative stress, and demyelination may further influence these relationships. Further studies investigating these dynamics, including imaging-based assessments in clinical and laboratory settings, are needed to better understand these pathways and to clarify how these diets may affect cognition.”
|
Round 2
Reviewer 1 Report
Comments and Suggestions for Authors
The 3rd version of the manuscript nutrients-3896917 entitled :"Mediterranean and MIND dietary patterns and cognitive performance in multiple sclerosis: a cross-sectional analysis of the UK Multiple Sclerosis Register", written by Yu and co-authors is ready for publication. This is an interesting and accurate cross-sectional investigation reporting association between targeted nutritional interventions and cognitive impairment in Multiple Sclerosis. The authors adopted a statistical model which also restrain the effect of outlier and erratic data collection, known to alter final results, while evaluating a number of affected subjects. Other statistical models could have been applied, but given the subject matter, the one used is also acceptable. The authors replied to all queries raised, providing a better explanation of their study design. The stratification by different MS conditions is anyway enough detailed due to the type of investigation. The results of this study are relevant even if modest and can be the starting point for longitudinal and interventional investigations. As well, they also encourage for prospective surveys combined with nutritional and biochemical analyses on the gut-brain axis molecular patterns as well as on inflammatory/immune parameters and redox substrates as indices of patients' endogenous state. The authors should maybe add this important sentence at the end of their paper.
Some other minor points:
- line 206: (<800 or >4,000 kcal/day for males; <500 or >3,500 kcal/day for females) : the authors probably intended "800-4,000 kcal/day for males; 500-3,500 kcal/day for females";
- Table 2: Ethnicity: the authors should avoid terms as White (European), but rather indicate British, Irish, other europeans; as well, the authors should not use the term black, but rather British African, Caribbean, African.
Author Response
Comment 1: " As well, they also encourage for prospective surveys combined with nutritional and biochemical analyses on the gut-brain axis molecular patterns as well as on inflammatory/immune parameters and redox substrates as indices of patients' endogenous state. The authors should maybe add this important sentence at the end of their paper."
Response 1: [Thank you for the constructive suggestion. We have added a sentence to the Conclusion highlighting the need for prospective and interventional studies incorporating gut–brain axis measures and inflammatory/immune and redox biomarkers to clarify elucidate mechanisms and test causality (p. 20, Line. 659).]
Comment 2: [line 206: (<800 or >4,000 kcal/day for males; <500 or >3,500 kcal/day for females) : the authors probably intended "800-4,000 kcal/day for males; 500-3,500 kcal/day for females"]
Response 2: [We thank the reviewer for noting this. We have corrected the energy-intake plausibility thresholds to read "800–4,000 kcal/day for males; 500–3,500 kcal/day for females" (line 206).]
Comment 3: [“Table 2: Ethnicity: avoid ‘White (European)’; indicate British, Irish, other Europeans. Do not use ‘black’; use British African, Caribbean, African.”]
Response 3: [Thank you for raising this important point. Ethnicity in our dataset was captured using the fixed response categories provided by the cognitive assessment questionnaire, which did not collect more granular national identities (e.g., British, Irish) or distinguish African/Caribbean subgroups. As such, we cannot accurately reclassify participants post hoc without misrepresentation.
Revisions made:
We have removed “(European)” and now report “White”.
Where compatible with the original item, we report “Black/African/Caribbean/Black British” rather than “Black” alone and have added a table footnote clarifying that category labels reflect the questionnaire options available and the granularity collected.
We will also align with the journal’s policies and preferred terminology during typesetting, provided this does not imply a level of detail not present in the source data.
Table 2 footnote:
“Ethnicity categories reflect the fixed response options in the source questionnaire; more granular national identities (e.g., British, Irish) and African/Caribbean subgroups were not collected.” ]